# MiniFold: Simple, Fast, and Accurate Protein Structure Prediction

## Abstract

Protein structure prediction has emerged as a powerful tool for biologists and drug makers. However, the computational toll associated with state-of-the-art models, such as AlphaFold2 or ESMFold, hinders their use in large-scale applications like virtual screening or mutational scanning, where a single experiment may involve processing millions of protein sequences. In an effort to develop a more efficient model, we aimed to understand which of the complex architectural choices proposed in AlphaFold2 were essential to achieve high performance, and which could be omitted without significantly compromising accuracy. This analysis culminated in a simple, yet highly expressive architecture for protein structure prediction. Our model, MiniFold, consists of a minimal Evoformer variant and a custom hardware-optimized implementation composed of newly designed GPU kernels. When compared against ESMFold, MiniFold achieves up to 20x speedup, up to 40x savings in memory, and shows improved scalability to long protein sequences while conserving over 97% of the original performance, making it a promising candidate for large-scale applications.

## 1 Introduction

With advances in deep learning based protein structure prediction, models such as Alphafold (Jumper et al., 2021) are now routinely used in biological discovery. Yet, the significant computational cost associated with these tools limits their use in large scale applications. As an illustration, consider the task of identifying disease relevant antibodies from patients' blood. A sequenced repertoire may contain millions if not billions of unique clonotypes Briney et al. (2019). At a speed of a few seconds per sequence, scanning 10 millions sequences would require over a GPU year.

Previous works have already recognized this challenge. To this end, they proposed to replace the multiple sequence alignment (MSA) stage of AlphaFold2 with a large protein language model (PLM), bringing structure prediction from minutes to seconds per sequence Lin et al. (2023); Wu et al. (2022). Yet, as highlighted above, even such speed can be severely limiting. Furthermore, this enhancement does not address the memory bottlenecks inherited by the complex Alphafold2 architecture. In order to obtain the desired efficiency, we must understand which parts of this architecture are crucial to performance, and which ones dominate the computation. The bulk of the compute is spent in the Evoformer blocks, which are responsible for predicting the pairwise interactions between the protein residues. Specifically, the Triangular attention layers of the Evoformer have $O(n^3)$ time and memory complexity, which hinders scaling to longer sequences.

Our goal, therefore, is to design a simple and efficient protein structure prediction model, without sacrificing accuracy. We present MiniFold, a highly optimized architecture for protein structure prediction which eliminates the structure module, and reduces the Evoformer layers to a single update using only triangular multiplicative operations and feed-forward layers. We demonstrate that MiniFold retains most of the model's expressive power and avoids the cubic memory blow-up associated with triangular attention. In addition, we introduce a hardware-optimized implementation of our model and devise novel GPU kernels to enhance both the throughput and memory efficiency of the model. Our proposed kernel optimization is applied to the self-gating operations in the triangular multiplicative updates and to the feed-forward networks.

We train MiniFold using the same protein language model used in ESMFold, and achieve up to 20x speedup compared to the original architecture and up to 40x savings in peak memory utilization, with improved scaling to longer protein sequences. On the standard CAMEO test set, MiniFold obtains a median lDDT of 0.88 compared to ESMFold's 0.90. On the more challenging CASP14 dataset, MiniFold obtains a median LDDT of 0.77 compared to ESMFold's 0.79, thus recovering over 97% of the original performance. Our work highlights the design choices that drive the success of protein structure prediction models, while enabling their use in high-throughput biological applications.

## 2 RELATED WORK

**Protein structure prediction**    Protein structure prediction has undergone a series of major break-throughs thanks to recent developments in deep learning methods. The current state-of-the-art model AlphaFold2 (Jumper et al., 2021) consists of two main components: a multiple sequence alignment (MSA) and template search, and a folding module responsible for decoding the MSA into a set of co-ordinates in 3-dimensional space. In particular, the derived MSA and templates are used to construct an initial set of sequence-level and pairwise-level representations. These are then fed to the folding module, more specifically to the Evoformer which learns to predict the pairwise distances between amino acid residues. Finally, a 3D-equivariant structure module predicts the 3D coordinate for each residue based on the encoded representation. RosettaFold (Baek et al., 2021) and UniFold (Li et al., 2022) adopts a similar MSA-based workflow but with a different model architecture. An important limitation of AlphaFold2, however, is its computational cost. Recently, protein folding models, such as OmegaFold (Wu et al., 2022) and ESMFold (Lin et al., 2023), accelerate inference by replacing the MSA database search with protein language models (Rives et al., 2021; Meier et al., 2021; Lin et al., 2023; Elnaggar et al., 2023; Chen et al., 2023). The input to these models require only single sequences, which is ideal for orphan proteins that do not have MSA. Different from PLM-based models that keeps the same architecture as AlphaFold2, we focus on improving the efficiency of the rest of model architecture to further improve its scalability.

**Efficiency & scalability**    Strategies to improve the efficiency of neural networks typically fall under one of four categories: pruning, quantization, GPU kernel optimization, and improvements in algorithmic complexity. Each of these approaches have been extensively studied in the context of the Transformer architecture (Vaswani et al., 2017). One such example is the recently proposed Flash-attention (Dao et al., 2022), which provides major speedup and memory savings to attention based architecture using an IO-aware implementation. Another example is Simple Recurrent Unit (Lei, 2021), which uses various GPU kernel optimization techniques for acceleration. Others such as Katharopoulos et al. (2020); Wang et al. (2020) propose algorithmic alterations that result in linear time complexity as a function of the input size. Yet, these methods are not obviously applicable to protein folding models, which utilize unique neural layers and transformations. Recognizing the importance of customized solutions to the model architecture at hand, our work aims to optimize protein structure prediction models to accelerate their inference speed, and improve their capacity to scale to long protein sequences. While recent work (Cheng et al., 2022; Wang et al., 2022) has begun to explore hardware-level optimization techniques to accelerate protein folding training, we focus on architecture-level optimization to improve inference speed.

## 3 MINIFOLD ARCHITECTURE

Our MiniFold model is composed of three modules: a protein language model, an efficient Evo-former (named MiniFormer), and a parameter-free coordinate realizer based on multi-dimensional scaling. Figure 1 illustrates the forward pass of our model and Figure 2 highlights the differences between MiniFold, ESMFold, and AlphaFold2 architecture.

The forward pass of MiniFold is summarized below. First, we encode the input protein sequence using the three-billion parameter ESM-2 language model  (Lin et al., 2023), which is a 36 layer Transformer. The embeddings at the last layer of ESM-2 are fed into a small feed-forward network and tiled into pairwise representations such that every entry $(i, j)$ is the concatenation of the vector representations of residues $i$ and $j$. Next, we feed the pair representation to the MiniFormer with 10 folding blocks, where the final block predicts a distogram matrix (pairwise distance between all residues). Finally, we pass the predicted distogram to the coordinate realizer and output the $C_\alpha$

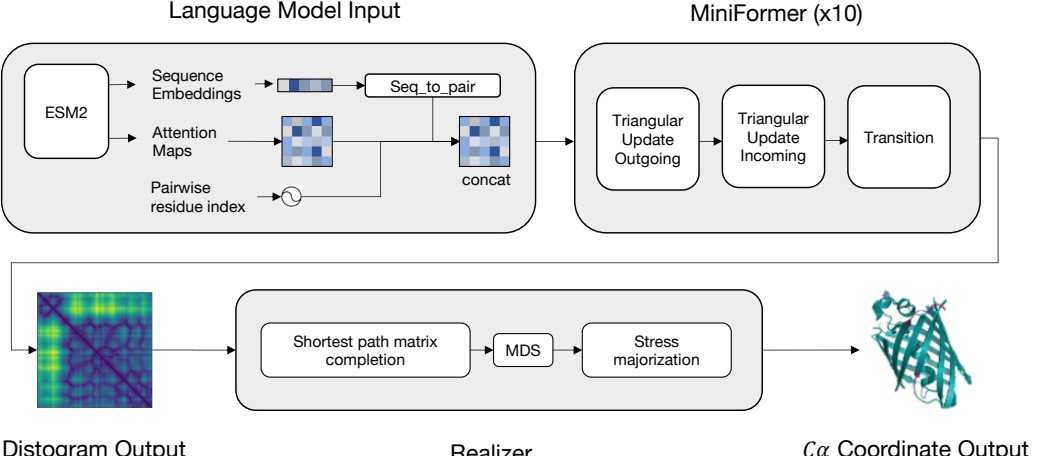

Figure 1: Our proposed MiniFold architecture. We use the ESM-2 protein language model to extract sequence level embeddings as well as pairwise attention maps. These are both concatenated and fed to the MiniFormer module which predicts the pairwise distances between the residues in the form of a binned distogram. Finally, to showcase the that the predicted distogram has indeed learned the 3-D structure of the protein, we use a ulti-dimensional scaling based realizer to output the $C_\alpha$ coordinates for each residue.

coordinates for each residue. In practice, for the large scale applications previously mentioned, the realizer may not be necessary but shows that the underlying structure learned by the distogram can indeed be realized in 3-D.

## 3.1 FROM EVOFORMER TO MINIFORMER

**Architecture changes** We perform several modifications to the Evoformer architecture, as shown in figure 2. First, we eliminate the sequence-level encoding and keep only the pairwise representation and update. Remarkably, this reduces the number of parameters from over 600 million to merely 3 million. In fact, we argue that the representational capacity of the Evoformer is influenced by the depth and complexity of its operations rather than by its parameter count.

Secondly, we remove the Triangular Attention block. There are two reasons for this change. First, this operation produces attention maps that consider every node triplet, which results in severe memory consumption. Secondly, we found that most of the expressive power of the Evoformer was driven by the triangular multiplicative blocks, which is considerably cheaper to compute. It is worth noting that the theoretical time complexity of the multiplicative update is also cubic in sequence length but the space complexity remains quadratic, thus resulting in a much cheaper operation.

The final block can be summarized by the following set of operations. Given some input tensor $\mathbf{X}$ of shape $(r, r, c)$, where $r$ is the number of residues and $c$ the number of channels:

$$\mathbf{X} = \mathbf{X} + \text{TriangularMultiplicativeUpdateOutgoing}(\mathbf{X})$$
$$\mathbf{X} = \mathbf{X} + \text{TriangularMultiplicativeUpdateIncoming}(\mathbf{X})$$
$$\mathbf{X} = \mathbf{X} + \text{FFN}(\mathbf{X})$$

In effect, the triangular updates serve as "token mixing" and the feed-forward network (or Transition layer) as "channel mixing".

**Efficient GPU Kernels** One of the key motivations for developing a minimal architecture was to identify the highly expressive components, and focus our performance engineering efforts. We set out to improve the efficiency of the triangular update blocks described above. In particular, we identified two promising areas of improvement: the self-gating operation in the multiplicative update, and the feed-forward layers where the hidden projection quadruples the input dimension.

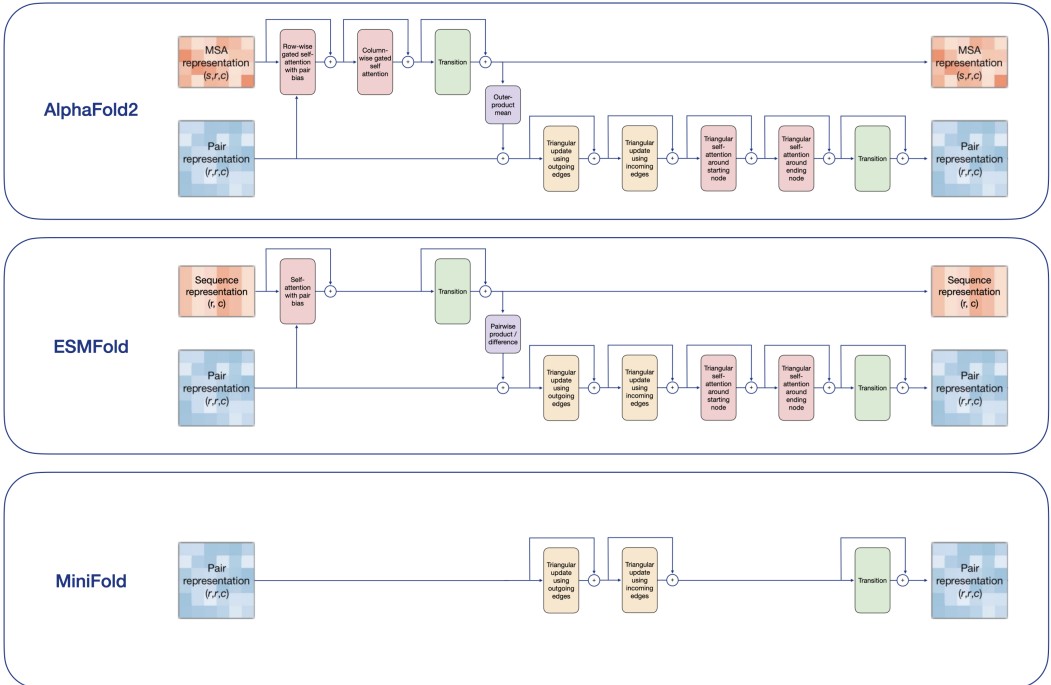

Figure 2: A single Evoformer block, from AlphaFold2, ESMFold and MiniFold. MiniFold removes the sequence based representation, as well as the triangular attention, reducing to the triangular multiplicative update and the transition feed-forward network. Images adapted from Lin et al. (2023) and Jumper et al. (2021).

SELF-GATING    The following gating operation appears twice in each update:

$$\boldsymbol{y} = (\boldsymbol{V}\boldsymbol{x} + \boldsymbol{b}_1) \cdot \text{sigmoid}(\boldsymbol{W}\boldsymbol{x} + \boldsymbol{b}_2)$$

Because all of these operations are applied point-wise, and $x$ is loaded twice, speedup and memory savings could be obtained by fusing this whole operation in a single kernel. This reduces the number of reads from 5 to 1, and the number of writes from 4 to 1, saving both memory and computing time. We provide pseudo-code for this formulation in Algorithm 2 in the appendix.

FEED-FORWARD    The feed-forward layer is ubiquitous in model deep learning architectures. It consists of an initial linear projection, followed by a non-linear activation, and another linear projection with a residual connection to the input. This layer is an essential part of the Transformer architecture Vaswani et al. (2017), as well as the folding blocks described above. This layer typically uses a hidden dimension that is larger than the input and output dimensions. In our case, it is 4x the input dimension. Our goal here is to avoid materializing the inner matrix, which can cause a memory bottleneck:

$$\boldsymbol{y} = \boldsymbol{x} + (\boldsymbol{W}_2\text{ReLU}(\boldsymbol{W}_1\boldsymbol{x} + \boldsymbol{b}_1) + \boldsymbol{b}_2)$$

Here, we observe that every row of the output $y$ requires precisely the same row in the input matrix $x$. Therefore, We can process each row in parallel, looping over the output dimension of $W_1$ and accumulating the result back into the input $x$. This allows us to never materialize the full inner matrix, resulting in important memory savings. As long as a row of the input matrix fits in a GPU thread, we only incur minimal extra reads of the weight matrices and bias vectors, but no extra computation. We provide pseudo-code for this formulation in Algorithm 3 in the appendix. The GPU kernel optimization is implemented using OpenAI's Triton package (Tillet et al., 2019).

---

**Algorithm 1:** Coordinate Realizer

---

**Input:** $logits \in \mathbb{R}^{N \times N \times 64}, d_{max} = 25\text{Å}, e \in \mathbb{R}^{64}$,
**Output:** $C_\alpha \in \mathbb{R}^{N \times 3}$

1   $D_{ij} \leftarrow \sum_{b=1}^{64} argmax((logits_{ij})_b \odot e_b), D_{ii} \leftarrow 0$

2   $W_{ij} \leftarrow \begin{cases} 1, & \text{if } (D_{ij} < d_{max}) \ \& \ (i \neq j) \\ 0, & \text{otherwise} \end{cases}$

3   $\boldsymbol{D} \leftarrow ShortestPath(\boldsymbol{D} \odot \boldsymbol{W})$

4   $J_{ij} \leftarrow \begin{cases} \frac{1}{N}, & \text{if } (i \neq j) \\ 0, & \text{otherwise} \end{cases}$

5   $\boldsymbol{B} \leftarrow -\frac{1}{2} * \boldsymbol{J} * \boldsymbol{D}^{\circ 2} * J$

6   $\Lambda, \boldsymbol{V} \leftarrow eigen\ decomposition(\boldsymbol{B})$

7   $\boldsymbol{C}_\alpha \leftarrow \boldsymbol{V} \odot \sqrt{\Lambda}$

8   **for** $(i = 0; i < 3; i = i + 3)$ **do**

9     |   $\boldsymbol{C}_\alpha \leftarrow \text{LBFGS}(\boldsymbol{D}, \boldsymbol{C}_\alpha)$

10 **end**

11 **return** $C_\alpha$

---

### 3.2 FROM STRUCTURE MODULE TO REALIZER

Alphafold2 introduced Invariant Point Attention (IPA) as a way to convert the output of the Evoformer into 3D coordinates. While this is an effective strategy, we wondered if coordinates could be inferred in a simple parameter-free fashion, directly from the predicted distogram. In particular, for the large scale applications mentioned such as virtual screening, it may not be necessary to compute the 3-D coordinates. Pariwsie distances are sufficient to identify stable folds, and discriminate for binders. Yet, in order to remove the structure module, we must ensure that it is not an important contributer to the final performance.

We propose a simple algorithm to realize coordinates. We first determine the predicted distance for each pair of residues by choosing the distance bin with the highest probability. Since the maximum distance bin is 25Å, we fill the missing entries by defining a graph with nodes as $C\alpha$ atoms and edges as distances under 25Å and running the all-pair shortest path algorithm on this graph. This results in an approximate distance for missing entries (i.e., pairs predicted to be farther than 25Å). Equipped with a full distance matrix, we perform classical multidimensional scaling (MDS) to generate initial coordinates for atoms. Because MDS (Mead, 1992) is sensitive to noise, the resulting coordinates generally do not satisfy the predicted distogram. Therefore, we refine the coordinates through three iterations of stress majorization with an LBFGS optimizer. We explored increasing the number of gradient descent stpes up to 20 but did not observe meaningful improvements. In fact, we find that this simple strategy is able to recover the coordinates and match the LDDT obtained from the predicted distances, which provides a useful upper bound on the reconstruction quality. Pseudo-code is provided in algorithm 1.

AlphaFold2 and ESMFold recycle the output of the structure module. In the absence of a structure module, we propose a change to the recycling strategy by directly recycling the argmax of the distogram to the MiniFormer. More spcifically, we convert the argmax back into a one-hot representation which we linearly project and sum into the input to the MiniFormer blocks. To further motivate this strategy, we "hack" ESMFold to utilize this approach instead of recycling the structure module. This analysis can be found in Table 1.

### 3.3 SIMPLIFIED TRAINING OBJECTIVE

AlphaFold and ESMFold were trained using a combination of multiple objective functions including distogram loss, FAPE loss, structural violation loss, MSA reconstruction loss, etc. When training MiniFold, we found it sufficient to using only the distogram loss to achieve high performance. Similar to previous work, we construct equally sized bins ranging from 2 to 25 Angstroms, and train the model to predict the pairwise distances between each pair of residues, using a classification objective over the bins described above, with the common cross entropy loss.

Table 1: Hacked ESMFold to recycle the distogram instead of the structure module's output. Results are comparable on CASP14. Note that these metrics are over C-beta atoms.

| Model | RMSD (↓) | TMScore (↑) | GDT-TS (↑) | lDDT (↑) |
|---|---|---|---|---|
| ESMFold (w/ structure recycling) | 4.81 | 0.76 | 0.65 | 0.75 |
| ESMFold (w/ distogram recycling) | 4.73 | 0.76 | 0.64 | 0.75 |

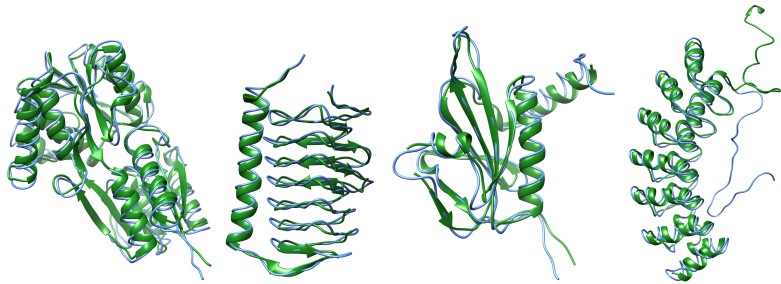

Figure 3: Comparison between structures predicted by MiniFold (blue) and ground truth in CAMEO (green). PDB ID from left to right: 7x0r, 7zwn, 7w52, 7xce.

## 4 EXPERIMENTS

In this section, we demonstrate the utility of our proposed approach by comparing the structure prediction accuracy, inference speed, and memory consumption against current state-of-the-art protein folding models including AlphaFold2, RosettaFold, ESMFold, and Omegafold (Jumper et al., 2021; Baek et al., 2021; Lin et al., 2023; Wu et al., 2022).

### 4.1 PROTEIN STRUCTURE PREDICTION ACCURACY

**Training and validation set**   We constructed a training set from the AlphaFold database. In particular, we first cluster the sequences from Uniref50 (Suzek et al., 2007) at 30% sequence similarity, and then select the structures with an average plDDT score above 0.7. This results in a high quality, diverse dataset of roughly 10 million structures. For simplicity, we opted to train on this dataset alone and do not use any experimental PDB structures. We randomly sample 10000 structure as our validation set for model selection and hyperparameter optimization.

**Test sets**   Following Jing et al. (2023), we evaluate MiniFold using CAMEO targets (Haas et al., 2019) released between Aug 1, 2022 and Oct 31, 2022, excluding any targets with more than 750 residues. The final test set consists of 183 structures. Similar to ESMFold, we also evaluate on 50 structures from the CASP14 targets, chosen to avoid overlaps with the ESMFold training set.

**Metrics**   To give a comprehensive measure of model performance, we follow previous work and report four metrics for each model: root mean square error (RMSD), template modeling score (TMScore), global distance test (GDT-TS), and local distance difference test (lDDT). Higher metrics correspond to better modeling performance, except for RMSD. All metrics are computed using the predicted $C_\alpha$ trace, and reported as the median over the test set.

**Hyperparameters and training details**   We train MiniFold using 10 layers of MiniFormer. We follow the parameters utilized in ESMFold with the exception of the hidden dimension used in the Triangular update, where we perform a downward projection from 128 to 32 dimension. We found that this projection had minimal effect on performance and provided meaningful computational savings. We train our model for 500K steps over roughly 3 days, on a single node using 8x A100 GPU's and a batch size of 16 per GPU for an effective batch size of 128 similar to ESMFold. We use the Adam optimizer with a learning rate of 1e-3 for all parameters, except the tuned language model layers for which we set a learning rate of 3e-5.

Table 2: Protein structure prediction results on CAMEO. All results are reported using the median performance over the $C_\alpha$ coordinates. We provide the MiniFold performance with and without recycling, with 3 recycling steps considered. MiniFold outperforms OmegaFold and RosettaFold while nearly matching the ESMFold performance.

| Model | RMSD (↓) | TMScore (↑) | GDT-TS (↑) | lDDT (↑) |
|---|---|---|---|---|
| AlphaFold2 | 1.64 | 0.95 | 0.91 | 0.93 |
| ESMFold | 2.03 | 0.93 | 0.88 | 0.90 |
| OmegaFold | 2.62 | 0.89 | 0.84 | 0.89 |
| RosettaFold | 3.17 | 0.84 | 0.75 | 0.82 |
| MiniFold (w/o recycling) | 2.85 | 0.90 | 0.84 | 0.87 |
| MiniFold (w/ recycling) | 2.62 | 0.91 | 0.86 | 0.88 |

Table 3: Protein structure prediction results on CASP14. All results are reported using the median performance over the $C_\alpha$ coordinates. We provide the MiniFold performance with and without recycling, with 3 recycling steps considered. MiniFold nearly matches ESMFold, but only with recycling enabled.

| Model | RMSD (↓) | TMScore (↑) | GDT-TS (↑) | lDDT (↑) |
|---|---|---|---|---|
| ESMFold | 4.57 | 0.77 | 0.67 | 0.79 |
| MiniFold (w/o recycling) | 8.19 | 0.66 | 0.59 | 0.73 |
| MiniFold (w/ recycling) | 5.53 | 0.74 | 0.65 | 0.77 |

### 4.1.1 RESULTS

**Accuracy** The performance on the CAMEO test set is described in Table 3.[1] We report the median performance for each metric. Our results show that MiniFold is competitive with other protein language model based structure prediction models, achieving over 97% of the state-of-the-art ESM-Fold at a fraction of the computational cost. As shown in Figure 3, the predicted structures by MiniFold align well with the ground truth on the test set, which further illustrates the effectiveness of our approach. Interestingly, we find that recylcing has limited effect on the CAMEO dataset but a very important one on the CASP14 targets. This suggests that for harder structures, recylcing enables more exploration of the energy landscape to find a good solution, but that it isn't always strictly necessary. It is worth noting that the performance of both ESMFold and MiniFold is considerably reduced on the CASP14 datasets. This is a known result, which is largly attributed to the language model being less effective than MSA-based approaches such as AlphaFold2. Future work may consider training an MSA-based version of MiniFold that can be more directly compared to AlphaFold2.

**Uncertainty estimation** Current protein folding models provide plDDT scores to quantify the uncertainty of predicted structures. We demonstrate that the entropy of the predicted distogram is a decent indicator of uncertainty. As shown in Figure 4, we find that predictions with lower entropy (e.g., higher certainty) tend to have higher LDDT scores. The Pearson correlation between true LDDT and distogram entropy is 0.60 on the CAMEO dataset and 0.9 on the CASP14 dataset. We also trained a pLDDT predictor, which obtains 0.76 correlation on CAMEO and 0.9 on CASP14. While not quite as effective as the pLDDT predictor, the distogram entropy is visibly a powerful measure of uncertainty.

---

[1]It is important to note that MDS is not chiral-aware and may produce mirror structures. Performance results as shown in table 3 assume the ideal chiral flip after $C_\alpha$ prediction.

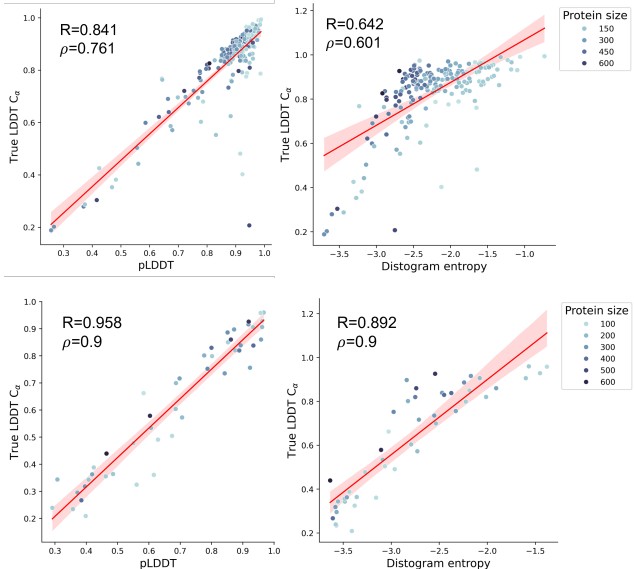

Figure 4: The trained pLDDT predictor (left) and the entropy over the predicted distogram (right) correlate with the true lDDT $C_\alpha$ as evaluated in the CAMEO dataset. CAMEO on top panel and CASP14 on bottom panel.

## 4.2 MINIFORMER EFFICIENCY

Next, we perform a systematic analysis of the throughput and memory consumption of our Mini-Former architecture and its corresponding hardware-optimized implementation. In particular we analyze the step by step progression from the Evoformer used in the ESMFold model to our proposed solution. Each of these steps were carefully chosen to reduce the computational cost, while minimizing any loss in predictive power.

- **Full**: This is the original Evoformer baseline in the ESMFold implementation, which uses the triangular updates from the OpenFold project (Ahdritz et al., 2022). The full Evoformer has 48 folding blocks and 3 recycling steps.
- **No sequence**: We remove the sequence track, keeping only the pairwise representation and its corresponding updates.
- **No attention**: We remove the triangular attention, keeping only the triangular multiplicative update.
- **Down projection**: We down-project the hidden embedding dimension in the triangular multiplicative blocks.
- **Kernels**: We include two highly optimized GPU implementations of our final architecture.
- **No recycle**: We remove the 3 recycles as used in ESMFold.
- **MiniFormer**: We include all optimization techniques and reduce the model to 10 layers with no recycling.

### 4.2.1 RESULTS

As shown in Figure 5, our results show that each of the steps proposed above contribute favorably to the model's improved efficiency. We note that the removal of the sequence track has minimal effect on inference speed, but reduces the number of trainable parameters considerably. The removal of the triangular attention, and the use of a downward projection in the multiplicative update yield substantial speedups, achieving near 5x improvement over the baseline. Our proposed kernels provide an another 2 to 3x speedup, leading to a total of 15 to 20x improvement in throughput for the full size model. Removing recycling and reducing the model size to 10 layers can yield up to 200x speedup.

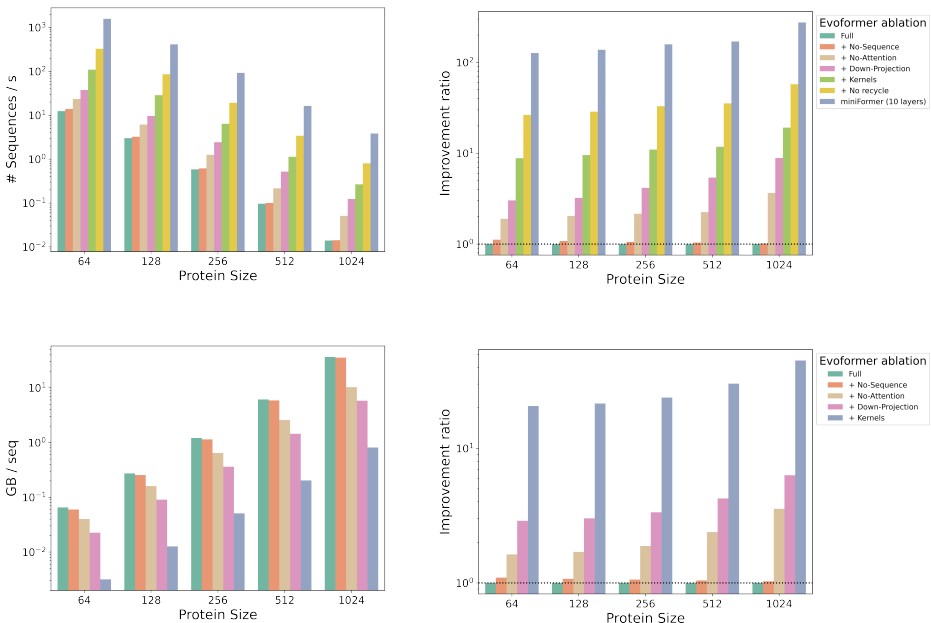

Figure 5: MiniFormer throughput analysis. The full model (blue bar) corresponds to the original Evoformer implementation in ESMFold. Our MiniFormer model (blue bar) with all optimization techniques (10 layers, removing sequence track, triangular attention, adding kernels) achieves 100-200x improvement in throughput over Evoformer. When using 40 layers and recycling, the model results in 15-20x speedup. The MiniFormer is also much more memory efficient with savings ranging from 20 to 40x depending on the sequence length.

The bottom panel of Figure 5 measures the peak memory usage during inference over different protein lengths. Here, we see that our proposed kernels result in substantial savings, and that MiniFold improves memory efficiency by nearly 20x for longer proteins. These results have important consequences regarding inference as well as training, allowing larger batch sizes which can result in faster training times.

## 5 CONCLUSION

In this work, we propose a highly efficient architecture for protein structure prediction, and a hardware-optimized implementation that results in considerable savings in both speed and memory while conserving most of its expressive power and performance. We also observe that a simple coordinate recovery algorithm based on multi-dimensional scaling can produce 3D coordinates from the predicted distogram with high accuracy. Our results have important implications regarding the use of protein structure prediction models for high throughput applications. In addition, we also provide a simple training scheme, using only a single objective function, dramatically simplifying the setup from previous work.

On the other hand, we observe that MiniFold still has room for improvement before matching the results of Alphafold2. In light of these results, we envision several areas of future work. First, we plan to bridge the performance gap by training MiniFold using multiple sequence alignments (MSA) as input, and to explore methods to speed up the construction, and utilization, of these MSAs. Second, we will focus on efficient full-atom reconstruction modules that can be added to the current architecture. Finally, we note that while our work builds from the success of PLM's in speeding up protein structure prediction, many of the optimizations shown in this work are not limited to PLM based methods and can be readily applied to MSA based models. We hope however that this work can serve as a stepping stone towards the ambitious goal of matching AlphaFold2's performance at a fraction of the cost.

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

## A APPENDIX

---

**Algorithm 2:** Self-gating kernel. Each parallel kernel thread $i$ is responsible for rows $R_i$ and columns $C_i$ of the output matrix $Y$

---

**Input:** input $X$, parameters $V$, $W$, output $Y$, kernel thread index $i$, block size $B$,
**Output:** $Y = \text{Linear}(V, X) \cdot \text{Sigmoid}(\text{Linear}(W, X))$

1   $X_i \leftarrow X[R_i, :])$               // Incurs one read operation for X
2   $V_i \leftarrow V[:, C_i])$
3   $W_i \leftarrow W[:, C_i])$
4   $Y_i^1 \leftarrow \text{DOT}(X_i, V_i)$
5   $Y_i^2 \leftarrow \text{DOT}(X_i, W_i)$
6   $Y_i \leftarrow \text{MUL}(\text{SIGMOID}(Y_i^1), Y_i^2)$
7   $Y[R_i, C_i] \leftarrow Y_i$            // Incurs one write operation for Y

---

---

**Algorithm 3:** Feed-forward kernel. Each parallel kernel thread $i$ is responsible for rows $R_i$ and all columns of the output matrix $Y$

---

**Input:** input $X$, parameters $V$, $W$, output $Y$, kernel thread index $i$, block size $B$,
**Output:** $Y = X + \text{Linear}(\text{ReLU}(\text{Linear}(X, V)), W)$

1   $X_i \leftarrow X[R_i, :])$              // Incurs one read operation for X
2   **for** $j \leftarrow 1$ *to* $N//B$ **do**
3      $V_j \leftarrow V[:, C_j])$         // Here $C_j$ is a block of B columns of V
4      $W_j \leftarrow W[C_j, :])$         // $C_j$ is also a block of B rows of W
5      $H \leftarrow \text{RELU}(\text{DOT}(X_i, V_j))$
6      $X_i \leftarrow X_i + \text{DOT}(H, W_j)$
7   **end**
8   $Y[R_i, :] \leftarrow X_i$           // Incurs one write operation for Y

---

