# OpenReview forum: "MiniFold: Simple, Fast and Accurate Protein Structure Prediction"
_ICLR.cc/2024/Conference — Submitted to ICLR 2024_

### Official Review · Reviewer_FoqV · 2023-10-19

**Soundness:** 1 poor
**Presentation:** 3 good
**Contribution:** 2 fair
**Rating:** 3
**Confidence:** 5

**Summary:**

AlphaFold2 remains the state of the art for protein structure prediction. However, the model has poor complexity characteristics and high peak memory usage, and inference on longer proteins often runs for several minutes or more. The authors propose MiniFold, a barebones, single-sequence architecture built with select modules from AlphaFold2. Despite the model's small parameter count, it achieves a high fraction of ESMFold's single-sequence performance on CAMEO proteins and runs faster.

**Strengths:**

The paper is clearly written and easy to follow. Some of the ablations are fairly surprising (e.g. recycling) and, if supported by better evaluations (see below), would improve our understanding of the all-important AlphaFold2 architecture. The proposed model is fast and fairly performant, and could make a useful addition to the structure prediction toolbox.

**Weaknesses:**

- Since MiniFold only predicts the backbone structure, the comparison to architectures like AlphaFold2 and ESMFold is a little bit unfair. It's unclear from this manuscript whether ablated components are actually unnecessary or necessary to predict the positions of atoms excluded here.
- There are a lot of important baselines missing here. RGN2 (Chowdhury et al., 2021) is a lightweight structure predictor with a language model that purports to be faster than AlphaFold2 by six orders of magnitude. Optimized versions of the full AlphaFold2 like FastFold (Cheng et al., 2022), UniFold (Li et al., 2022), and recent versions of OpenFold (Ahdritz et al., 2022) also exist but are not tested here. RosettaFold2 is also missing.
- CAMEO evaluation is not sufficient to tease out the differences between structure prediction models. The gold standard is CASP proteins, which often reveal much larger gaps between models than can be seen in CAMEO figures alone. ESMFold famously underperformed AlphaFold2 at CASP15 by a very wide margin, and I suspect the limited capacity of MiniFold could hold it back even further here.
- Originality isn't the only criterion here, but I'm not sure if this paper has many new insights that would be of interest to the broader machine learning community. The observation that triangle attention isn't strictly necessary was already noted in the original AlphaFold2 paper. Other more surprising claims (recycling isn't necessary) need to be supported by additional evidence, like a CASP evaluation. As I mentioned above, this isn't the first paper to present an optimized, protein-language-model based lightweight alternative to AlphaFold2 either.

Bits and bobs:

> Our analysis reveals that the bulk of the compute is spent in the Evoformer blocks, which is responsible for predicting the pairwise interactions between the protein residues. Specifically, the Triangular attention layers of the Evoformer have O(n3) time and memory complexity, which hinders scaling to longer sequences.

- This is a bit grandiose. The complexities of the AlphaFold2 modules are already well-known.

> The main limitation of AlphaFold2, however, is its computational cost

- I wouldn't call this the "main" limitation. The model isn't that great at proteins without deep MSAs, e.g..

>We constructed a training set from the AlphaFold database. In particular, we first cluster the sequences from Uniref50 (Suzek et al., 2007) at 30% sequence similarity, and then select the structures with an average plDDT score above 0.7. This results in a high quality, diverse dataset of roughly 10 million structures.

- If you're training on AF2 structures, especially exclusively, then a lot of your claims about certain components of AF2 being unnecessary could be called into question; this is effectively a form of distillation, not an independent competing architecture.

>Our results show that MiniFold is competitive with other protein language model based structure prediction models, achieving over 95% of the state-of-the-art ESMFold at a fraction of the computational cost.

- 95% is somewhat misleading. OpenFold shows that the full AlphaFold2 model reaches comparable lDDT levels almost immediately; almost all of the training time is spent closing roughly the same gap as the one between MiniFold and the full AlphaFold2 model.

**Questions:**

>We perform several modifications to the Evoformer architecture, as shown in figure 2. First, we eliminate the sequence-level encoding and keep only the pairwise representation and update. Remarkably, this reduces the number of parameters from over 600 million to merely 3 million. In fact, we argue that the representational capacity of the Evoformer is influenced by the depth and complexity of its operations rather than by its parameter count.

- Where does 600 million come from? AlphaFold2 has about 93 million parameters.

>In addition, we eliminate the recycling operation used in both AlphaFold2 and ESMFold, as we found it only provides very minimal benefits that did not outweigh the additional computational cost.

- This contradicts the ablations in the AlphaFold2 paper. In what sense does it only provide minimal benefits? On which sequences?

---

> ### Author Response · Authors · 2023-11-23
>
> We would like to thank the reviewer for their comments, which helped us improve the paper! Please find our response below:
>
> > Since MiniFold only predicts the backbone structure, the comparison to architectures like AlphaFold2 and ESMFold is a little bit unfair. It's unclear from this manuscript whether ablated components are actually unnecessary or necessary to predict the positions of atoms excluded here.
>
> This is certainly a fair point. However, the no-IPA ablation in the AlphaFold2 paper seems to suggest that removing the structure module does not have a significant impact on performance. Therefore we opted to remove it to gain further speed. Yet, for this to make sense, we also needed to demonstrate that the MiniFormer had sufficiently learned the underlying 3D structure, which the coordinate realizer shows. The RGN2 model did not predict full-atom either, and calls the AlphaFold2 refinement code to do so.
> > There are a lot of important baselines missing here. RGN2 (Chowdhury et al., 2021) is a lightweight structure predictor with a language model that purports to be faster than AlphaFold2 by six orders of magnitude. Optimized versions of the full AlphaFold2 like FastFold (Cheng et al., 2022), UniFold (Li et al., 2022), and recent versions of OpenFold (Ahdritz et al., 2022) also exist but are not tested here. RosettaFold2 is also missing.
>
> Thank you for pointing this out. Omitting these models was intentional for the following reasons:
>  RGN2 does not perform well on structure prediction and was proposed in the context of design and orphan proteins, at a time where AlphaFold2 did not have good support for single sequence prediction. A recent publication benchmarking on CASP shows this: https://www.sciencedirect.com/science/article/pii/S0959440X2300101X. In figure 2 it is clear that RGN2 severely underperforms compared to OmegaFold and ESMFold.
> ESMFold is built on top of OpenFold. Thus when benchmarking speed, we directly compared to Openfold’s implementations. The OpenFold repository also mentions that FastFold’s implementations have been merged in, and focused largely on the attention modules, which we removed in this work. Furthermore, the speed-ups shown in these repositories are much smaller than the ones we report, while the memory savings involve a chunking strategy which reduces memory at the cost of speed, while our approach results in memory savings that also produce speedup.
> RosettaFold2 is largely similar to AlphaFold2, and includes a structure track using the expensive SE(3) Transformer. We did not think it added a meaningfully different data point beyond AlphaFold2.
>
> > CAMEO evaluation is not sufficient to tease out the differences between structure prediction models. The gold standard is CASP proteins, which often reveal much larger gaps between models than can be seen in CAMEO figures alone. ESMFold famously underperformed AlphaFold2 at CASP15 by a very wide margin, and I suspect the limited capacity of MiniFold could hold it back even further here.
>
> Thank you for this suggestion. We totally agree with this comment, and have produced thorough experiments on CASP14 (see global answer). This analysis shows that we remain very competitive with ESMFold but only with the 40 layer model, and with recycling enabled. In this setting, our speedup is reduced to up to 20x, but our memory savings of up to 40x remain, which is still significant. We have modified the paper abstract, and included these new results.

---

> ### Author Response · Authors · 2023-11-23
>
> > Originality isn't the only criterion here, but I'm not sure if this paper has many new insights that would be of interest to the broader machine learning community. The observation that triangle attention isn't strictly necessary was already noted in the original AlphaFold2 paper. Other more surprising claims (recycling isn't necessary) need to be supported by additional evidence, like a CASP evaluation. As I mentioned above, this isn't the first paper to present an optimized, protein-language-model based lightweight alternative to AlphaFold2 either.
>
> We think there are several important contributions in our work that distinguishes it from previous work and that would be of interest to the machine learning community:
>
> 1. First, we produced a model that is very close to ESMFold (<3% drop) and results in massive speedup and memory savings. This could enable many more in the research community being able to work on the protein structure prediction task, especially when only limited compute is available. While ESMFold was a great first step in this direction, it remained very expensive to train from scratch or fine-tune.
>
> 2. Our proposed kernels are broadly applicable, beyond this architecture.  In particular, the memory efficient 2 layer feed-forward could be applied to any Transformer model.
>
> 3. We contribute several analyses that provide better understanding of the AlphaFold2 architecture. The first one is to show that the distogram alone, without any other learned projection, has learned the 3D structure, which shows that the structure module is largely a decoding mechanism. We also showed that the uncertainty of the distogram is a great predictor of structure accuracy. Finally, we showed that the sequence track can be completely eliminated in ESMFold, reducing the parameter count to just a few million parameters. This result emphasizes that the power of the architecture lies in the expressivity of its operations and not in its parameter count.
>
> 4. We obtained a powerful model by using just a single loss function, and a single stage of training.
>
> Regarding the question of recycling, we acknowledge that our statements were misleading based on CAMEO results. CASP14 results clearly show the importance of recycling. We thank the reviewer for flagging this, and have modified the paper accordingly.
>
> >This is a bit grandiose. The complexities of the AlphaFold2 modules are already well-known.
>
> This was not our intention, we have modified the text. On the other hand, some reviewers commented on the ESM language model being heavy, even though it only covers 1% of the compute time. We do believe that this is indeed a counter-intuitive piece of information given the very large number of parameters in the language model (3 billion).
>
> > I wouldn't call this the "main" limitation. The model isn't that great at proteins without deep MSAs, e.g..
>
> This is totally fair, we have changed the text.
>
> > If you're training on AF2 structures, especially exclusively, then a lot of your claims about certain components of AF2 being unnecessary could be called into question; this is effectively a form of distillation, not an independent competing architecture.
>
> We did not intend to say that these were completely unnecessary, but rather that thanks to this distillation setting, it may be possible to simplify the architecture, which seems to be the case. In fact, our main objective was to speed-up ESMFold, which also utilizes AlphaFold2 structures for training.
>
> > 95% is somewhat misleading. OpenFold shows that the full AlphaFold2 model reaches comparable lDDT levels almost immediately; almost all of the training time is spent closing roughly the same gap as the one between MiniFold and the full AlphaFold2 model.
>
> We have trained a larger model which has further reduced the gap to < 3% but agree that these last few percent are not negligible. Yet we also do not want to trivialize this result. Reaching this performance required careful experiments to identify opportunities for speed-up that did not result in very large performance drops.
>
> > Where does 600 million come from? AlphaFold2 has about 93 million parameters.
>
> The Evoformer in ESMFold uses different hyperparameters, resulting in > 600M parameters in the folding trunk.
>
> > This contradicts the ablations in the AlphaFold2 paper. In what sense does it only provide minimal benefits? On which sequences?
>
> As mentioned previously, this was indeed a mistake on our part. Recycling is important on CASP14. We have modified the text accordingly. We did make an interesting observation that recycling the distogram is sufficient to recover the performance drop compared to ESMFold.

---

### Official Review · Reviewer_mayn · 2023-10-30

**Soundness:** 3 good
**Presentation:** 3 good
**Contribution:** 2 fair
**Rating:** 5
**Confidence:** 3

**Summary:**

This paper proposes an efficient protein structure prediction method. The authors use ESM-2 to extract residue features and construct pairwise representations. The pairwise features are used to predict pairwise distance between all residues. Finally, they recover  3D coordinates from
the predicted distogram based on multi-dimensional scaling.

**Strengths:**

1. The paper is clearly written.
2. The main strength of MiniFold is good efficiency. The proposed MiniFold achieves over 100x speedup.
3. The authors simplified the modeling complexity and did GPU kernel optimization.

**Weaknesses:**

1. The proposed methods only generate C-alpha atoms. It could not reconstruct full-atoms.
2. Most of the neural modules were copied directly from existing literature, limiting the novelty.
3. The authors do not provide code for checking.

**Questions:**

1. Could the author provide TMScore comparisons against AlphaFold, Omegafold, and ESMFold on long proteins (protein size>1000)?

2. Have you tried to reconstruct full-atoms?

3. Considering the importance of the structural prediction task, it should be carefully examined. Could the authors provide the source code?

4. Does the inference time cost take into account the overhead of the coordinate realization module?

---

> ### Author Response · Authors · 2023-11-23
>
> We would like to thank the reviewer for their comments. Please find the answer to your points below.
>
> > Most of the neural modules were copied directly from existing literature, limiting the novelty.
>
> We think there are several important contributions in our work that distinguishes it from previous work and that would be of interest to the machine learning community:
>
> 1. First, we produced a model that is very close to ESMFold (<3% drop) and results in massive speedup and memory savings. This could enable many more in the research community being able to work on the protein structure prediction task, especially when only limited compute is available. While ESMFold was a great first step in this direction, it remained very expensive to train from scratch or fine-tune.
>
> 2. Our proposed kernels are broadly applicable, beyond this architecture.  In particular, the memory efficient 2 layer feed-forward could be applied to any Transformer model.
>
> 3. We contribute several analyses that provide better understanding of the AlphaFold2 architecture. The first one is to show that the distogram alone, without any other learned projection, has learned the 3D structure, which shows that the structure module is largely a decoding mechanism. We also showed that the uncertainty of the distogram is a great predictor of structure accuracy. Finally, we showed that the sequence track can be completely eliminated in ESMFold, reducing the parameter count to just a few million parameters. This result emphasizes that the power of the architecture lies in the expressivity of its operations and not in its parameter count.
>
> 4. We obtained a powerful model by using just a single loss function, and a single stage of training.
>
>
> > Could the author provide TMScore comparisons against AlphaFold, Omegafold, and ESMFold on long proteins (protein size>1000)?
>
> We did not get the opportunity to do so in the limited time available but will aim to include these results in the camera ready.
>
> > Have you tried to reconstruct full-atoms?
>
> There are a variety of deep learning based methods for predicting all-atom structures from C alpha atoms, such as DLPacker (Misiura et al. 2021). These methods are computationally efficient and reasonably accurate. We can use these methods to predict full-atom structure given C alpha atom coordinates predicted by MiniFold. As a side note, RGN2 also uses external software to predict full-atom structure since it only predicts C alpha atom coordinates.
>
> Misiura et al., DLPacker: Deep Learning for Prediction of Amino Acid Side Chain Conformations in Proteins. Proteins, 2021
>
> > Considering the importance of the structural prediction task, it should be carefully examined. Could the authors provide the source code?
>
> We will release the code upon publication.
>
> > Does the inference time cost take into account the overhead of the coordinate realization module?
>
> It does not. The main reason here is that we argue that for the large-scale applications of interest such as virtual screening, full coordinate recovery may not be important. Yet, to make this point, it’s important to know if the output of the evoformer has already learned the structure and that the structure module is mostly decoding this information rather than making a significant contribution to performance. We have added an analysis using the ESMFold distogram which confirms this result. One could also train a small structure module or a single projection from the Evoformer embeddings similar to what is proposed in the AlphaFold2 ablation studies.

---

### Official Review · Reviewer_cGwq · 2023-10-31

**Soundness:** 3 good
**Presentation:** 4 excellent
**Contribution:** 3 good
**Rating:** 6
**Confidence:** 4

**Summary:**

This paper tackles the protein folding problem, an important and time-consuming task. The authors proposed MiniFold that can infer a structure with 100x acceleration and tolerable accuracy loss. To achieve this, they carefully studied each block in EvoFormer, removed unnecessary blocks, and proposed a new pipeline. Experimental results demonstrate the effectiveness of their algorithm.

**Strengths:**

Overall, I pretty much enjoy reading this paper. The motivation and intuition are clear, and the way the authors solve the problems is reasonable. In addition, the contribution is indeed significant, especially under large-scale screening demand. The paper is well-written and organized, and the demonstration is straightforward.

**Weaknesses:**

Currently, I give a score of 6. I am happy to increase my score if the below weaknesses (and questions) can be appropriately addressed.

-  First of all, the authors clearly have dived into the (time and performance) ablation of AlphaFold2, ESMFold and MiniFold, which will be great to present. For example, how do time and performance change if we remove the triangular attention blocks? How is the performance change if we recycle MiniFold once / twice / third times?Theses results are not only useful for the design of MiniFold, but are also knowledge that people are curious about.
- The second question relates to the third part of MiniFold (structural determination based on distance matrix). Why can't you directly build the 3D $C_\alpha$ backbone based on the distance matrix (assuming that the matrix is filled)? In addition, how do you determine the side-chain angle?
- Third, I noticed that you built the model on a smaller (but pre-trained) ESMFold, right? In fact, one of the reasons why ESM and AF2 are so huge is that they train *from scratch*. I am not saying that using previous embeddings is not good, but an intermediate path to address the efficiency problems is to distill or prune the ESM model, or to use it as a teacher model to train a smaller model (since you have an infinitely large dataset now). Do you know any work about how this would be compared to MiniFold? What do you think is the pros and cons of different solution paths?
- A minor point, I don't think "removing" cycling can be counted as your acceleration. This is entirely a time/accuracy trade-off, and can be easily achieved. I'd suggested the authors cycle MiniFold as well and compare the time.
- Last but I am indeed curious: you mention that the Multiplicative layer can replace attention. Can you show how the performance changes after the replacement? In addition, this is also applicable to other fields, so I am curious why do you think this replacement can work well, and is it a domain-specific thing or not?

**Questions:**

I have asked many questions above.

---

> ### Author Response · Authors · 2023-11-23
>
> We would like to thank the reviewer for their comments. Please find our answers to your points below:
>
> > First of all, the authors clearly have dived into the (time and performance) ablation of AlphaFold2, ESMFold and MiniFold, which will be great to present. For example, how do time and performance change if we remove the triangular attention blocks? How is the performance change if we recycle MiniFold once / twice / third times?Theses results are not only useful for the design of MiniFold, but are also knowledge that people are curious about.
>
> Running full ablations using the Triangular Attention blocks is difficult as the memory requirements get so large that we could not train more than a very small number of layers with it. This makes it difficult to compare with full sized models. This being said, in our early experiments on small models we observed very little loss in performance when omitting the triangular attention, so long as we kept the triangular multiplicative update. Furthermore, our results show that we are able to nearly match ESMFold without it, which is another positive signal.
>
> > The second question relates to the third part of MiniFold (structural determination based on distance matrix). Why can't you directly build the 3D backbone based on the distance matrix (assuming that the matrix is filled)? In addition, how do you determine the side-chain angle?
>
> This is precisely the issue, the matrix is not filled. The model only predicts distances up to 25 A, with the last bin covering all distances larger than 25. Therefore we first need to approximate the missing distances, which we do using the all pairs shortest path algorithm. Once that is done, we can use MDS to get the coordinates. However because of the approximation of long distances, MDS will not be precise so we need a few steps of gradient descent to fix it.
>
> > Third, I noticed that you built the model on a smaller (but pre-trained) ESMFold, right? In fact, one of the reasons why ESM and AF2 are so huge is that they train from scratch. I am not saying that using previous embeddings is not good, but an intermediate path to address the efficiency problems is to distill or prune the ESM model, or to use it as a teacher model to train a smaller model (since you have an infinitely large dataset now). Do you know any work about how this would be compared to MiniFold? What do you think is the pros and cons of different solution paths?
>
> ESMFold is not trained from scratch. The ESM-2 language model is trained first, and then ESMFold is trained on top of the ESM language model. Here similarly we assume that we also have the ESM2 language model and train the folding model from scratch. Furthermore, the ESM2 language model is very cheap compared to the Evoformer (only 1% of total compute time) so we do not believe it needs further optimization.
> Pruning ESMFold is an interesting idea but difficult to do as it would likely require fine-tuning the model through the pruning process. Furthermore, unstructured pruning leads to smaller model footprint but rarely improves speed. Structured pruning approaches may be utilized but tend to not perform as well. Here we take a simpler route, by identifying parts of the architecture that are not the main driver of performance, and successfully training a streamlined model from scratch.
> Regarding student-teacher, note that we are using AF2 predicted structures for training, which is arguably already a student-teacher setting.
>
> > A minor point, I don't think "removing" cycling can be counted as your acceleration. This is entirely a time/accuracy trade-off, and can be easily achieved. I'd suggested the authors cycle MiniFold as well and compare the time.
>
> We did not intend to make it one of our contributions, but to highlight the extra savings from removing recycling. We fully agree that an ablation including recycled manifold is important and have added this to the paper. The results show that recycling is important on CASP14 targets but not on the CAMEO test set.
>
> > Last but I am indeed curious: you mention that the Multiplicative layer can replace attention. Can you show how the performance changes after the replacement? In addition, this is also applicable to other fields, so I am curious why do you think this replacement can work well, and is it a domain-specific thing or not?
>
> It’s not so much a replacement since both the triangular attention and triangular multiplicative updates are present in the original architecture. In our early experiments, we found that the triangular multiplicative seemed to perform well even in the absence of the triangular attention. The triangular attention being very computationally heavy, we opted to remove it.

---

### Official Review · Reviewer_NKzF · 2023-10-31

**Soundness:** 3 good
**Presentation:** 3 good
**Contribution:** 3 good
**Rating:** 3
**Confidence:** 4

**Summary:**

The paper presents MiniFold, a highly optimized protein structure prediction model that achieves over 100x speedup compared to ESMFold while retaining 95% of its accuracy. MiniFold simplifies the Evoformer architecture by removing unnecessary components like sequence encoding, triangular attention, and recycling. It also implements efficient GPU kernels and a simple coordinate recovery module.

**Strengths:**

- The analysis to identify key components of Evoformer that enable high performance protein folding is insightful. This allows simplifying the architecture while maintaining accuracy.
- The 100x speedup over ESMFold enables the application of protein folding models to high-throughput tasks involving millions of sequences. This is a major contribution.

**Weaknesses:**

- The prediction process is faster, but final performance significantly decreases.
- Removing IPA is disadvantageous, as the structure module is less costly than Evoformer.
- Kernels are implemented with OpenAI's Triton, not CUDA; a full-page explanation is unnecessary due to well-known engineering improvements.
- The analysis of kernels is wrong. For example, "This reduces the number of reads from 5 to 1, and the number of writes from 4 to 1". The Wx and Vx are matrix multiplication operators, which will call GEMM kernels, thus these read/write cannot be merged. We usually can only save the read/write times for element-wise operators.
- The method relies on a computationally demanding pretrained protein language model; simplification would be beneficial.
- Coordinate recovery omits chirality consideration, potentially negatively impacting performance.
- In-depth analysis of uncertainty estimation technique is needed for better understanding of robustness.

**Questions:**

- I notice there are confidence scores, but where you do inject randomness to generate a distribution?
- Could training with MSAs further improve MiniFold's accuracy? What optimizations would be needed?
- In Sec 4.2, how do you perform MiniFormer with recycling based on MDS?
- Do you end-to-end optimze the coordinates with gradient back-propagation, or just the Distogram?
- Do you include the time cost of ESM when compute time cost in Sec 4.2?
- Does the comparison in Sec 4.2 include the gradient backward time?
- Are there standalone efficiency comparsion (compared with the ones without kernels) for the two optimized kerenls?

---

> ### Author Response · Authors · 2023-11-23
>
> We would like to thank the reviewer for their comments. We have addressed each point below.
>
> > The prediction process is faster, but final performance significantly decreases.
>
> We agree that there is a performance decrease compared to ESMFold, but it’s important to note that the model we used only had 10 layers and was therefore at a disadvantage compared to ESMFold’s 48 layers. We have added a 40 layer model for a closer comparison. This model results in an improvement in performance which reduces the gap to under 3%. Longer training, and matching the full 48 layer model should bring the numbers very close, which we plan on exploring. Comparisons to the AlphaFold2 numbers are less meaningful as they utilize MSA’s as input, and we do not.
>
> > Removing IPA is disadvantageous, as the structure module is less costly than Evoformer.
>
> For the applications mentioned (virtual screening), it’s not clear that obtaining the exact coordinates is necessary. Thus the question we posit is whether the structure module improves performance or is rather decoding the distogram from the Evoformer. This is in line with the ablation studies in the AlphaFold2 paper which showed that replacing the IPA with a single projection was sufficient for decoding (see section 1.13 in the AlphaFold2 supplementary material). Here we wondered if we would lose performance by removing such projection too, and our results show that coordinate recovery from predicted distograms is very effective. This realization strategy is not meant to be used in practice. Instead it is used to show that the distogram has indeed learned a valid structure.
>
> > Kernels are implemented with OpenAI's Triton, not CUDA; a full-page explanation is unnecessary due to well-known engineering improvements.
>
> We felt it was important to highlight how we used triton to achieve this, as it involved important design decisions that are non-trivial, and beyond the use of the Triton language. This is particularly relevant for the feed-forward kernel which could be implemented in several different ways, with different trade-offs. This being said, we agree that some of the details of the analysis could be moved to the appendix, which we did. Thank you!
>
> > The analysis of kernels is wrong. For example, "This reduces the number of reads from 5 to 1, and the number of writes from 4 to 1". The Wx and Vx are matrix multiplication operators, which will call GEMM kernels, thus these read/write cannot be merged. We usually can only save the read/write times for element-wise operators.
>
> The part that only requires a single read and write is meant to refer to entries of the input x which has shape B x N x N x D, where B is the batch size, N the sequence length and D the hidden dimension. The input is what dominates the memory usage and need only be read once and written back once per B x N x N vector of dimension D. Reads and writes can be merged because the operations are in fact elementwise with respect to the first three dimensions, so we are computing Wx_bij and Vx_bij and apply the subsequent operations while Wx_bij and Vx_bij are still in SRAM. We’re not sure which GEMM kernels the reviewer is referring to here, and how they affect this analysis.  Based on our design, Triton generates a custom kernel, which is able to load data just once. The fact that we are doing matrix multiplication within the kernel does not make it a standard GEMM kernel. Please let us know if we missed part of the point being made here.
>
> > The method relies on a computationally demanding pretrained protein language model; simplification would be beneficial.
>
> It turns out that the throughput of the language model, despite its size, is excellent. In fact, when looking at throughput the language model only covers 1% of the total compute time, compared to 82% for the Evoformer and 17% for the structure module at a sequence length of 256. Thus we did not see any reason to focus our efforts on the language model.
>
> > Coordinate recovery omits chirality consideration, potentially negatively impacting performance.
>
> We trained a simple chirality flipped predictor using just the torsion angle along the C-alpha trace, and found that we could identify full flips with near perfect accuracy. The intuition here is that alpha helices are almost always right-handed, which is easily detectable.
>
> > In-depth analysis of uncertainty estimation technique is needed for better understanding of robustness.
>
> We have added more results on the uncertainty measurements, including a comparison between distogram entropy and a pLDDT predictor (see figure 4).
>
> > I notice there are confidence scores, but where you do inject randomness to generate a distribution?
>
> We do not inject any randomness. Since the distance prediction task is a classification over bins, we can compute a categorical distribution over the bins. The confidence score is then the negative entropy of this distribution averaged over every pair.

---

> > ### Author Response · Authors · 2023-11-23
> >
> > > Could training with MSAs further improve MiniFold's accuracy? What optimizations would be needed?
> >
> > Certainly, and this is a very important area of future work. We believe that the observations in this work will translate to AlphaFold2 as well. One main point that will need to be ironed out is whether the sequence track is more critical in AF2 than it is in ESMFold. In this work we aimed to show that this simplified architecture was just as powerful as the full ESMFold architecture.
> >
> > > In Sec 4.2, how do you perform MiniFormer with recycling based on MDS?
> >
> > It is not necessary to compute MDS for recycling, one can simply feed the distogram back into the Evoformer. We have added results using this recycling and show that it provides similar improvements than the ones observed in ESMFold (see table in our overall response). We also ran an experiment by hacking ESMFold to recycle using the argmax of its distogram instead of its structure module. This experiment showed that recycling the distogram is just as effective as recycling with the structure module output.
> >
> > > Do you end-to-end optimze the coordinates with gradient back-propagation, or just the Distogram?
> >
> > No end-to-end on coordinate optimization. It appears that the distogram has learned enough to reconstruct the coordinates.
> >
> > > Do you include the time cost of ESM when compute time cost in Sec 4.2?
> >
> > No, the analysis is only on the Evoformer. This is because, as mentioned above, the ESM language model only covers 1% of the computation cost, which is close to negligible.
> >
> > > Does the comparison in Sec 4.2 include the gradient backward time?
> >
> > It does not, these are inference measurements.
> >
> > > Are there standalone efficiency comparsion (compared with the ones without kernels) for the two optimized kernels?
> >
> > The self-gating kernel improves this operation by 2-3x and saves 3x memory. The FFN kernel provides 1.5-2x speedup with an 8x memory saving when the inner-dimension is increased by a factor of 4, which is typical in the Transformer architecture and is the hyperparameter value used by AlphaFold2 and ESMFold.

---

> ### Comment · Reviewer_NKzF · 2023-11-23
> **Response to the Author Rebuttal**
>
> In the discussion about kernel analysis, it appears that the authors may not have extensive expertise in high-performance computing. Regardless of the method employed, be it Triton or CUDA, merely combining everything into a single kernel does not ensure that the operators are effectively fused to minimize the intermediate global memory write/read costs. Kernel fusion is typically applied to element-wise operators, which are constrained by memory bandwidth rather than computational capacity. Conversely, matrix multiplications are computationally intensive and generally utilize highly-optimized GEMM kernels to achieve peak performance. Fusing these GEMM kernels with element-wise operators and saving the intermediate memory cost is a complex task, even for a CUDA expert. This task's complexity is amplified in automated translation scenarios with Triton. If the authors themselves struggle with manual kernel optimization, relying on Triton for automatic, efficient kernel fusion seems overly optimistic, casting doubt on the credibility of their analysis results.

---

### Author Response · Authors · 2023-11-23
**Improved results and analysis**

We would like to thank all reviewers for their thoughtful comments, many of which helped us better understand the capacity of our proposed model. There are a few points which we wish to address globally:

1. As recommended, we have run extensive experiments on CASP14, and found that our model’s performance remained close to the ESMFold performance but only when enabling recycling. It’s worth noting that recycling has a minimal effect on CAMEO. Results have been added to the paper, and are displayed in the table below.

2. We trained a larger 40 layer model, which reduces the gap to ESMFold but reduces our speedup. When using the 40 layer model with recycling, we achieve a 20-30x speedup compared to ESMFold, which is still consequential. Our memory savings are not affected by this change, and remain at 20x. Results have been added to the paper, and are displayed in the table below. We have modified the text to reflect the speedup under this setting, since it was critical to having comparable performance to ESMFold on CASP14.

3. We propose a variant of recycling that operates uniquely within the Evoformer (without a structure module), and show that it is effective on CASP14. To further validate this observation, we hacked ESMFold to utilize the argmax of the distogram instead of the structure module’s output for recycling. Again, we observed a similar result that this recycling strategy is just as effective.

4. We expanded on the uncertainty analysis by comparing the distogram uncertainty to a plDDT predictor.


| Model | RMSD | TMScore | GDT-TS | lDDT |
|----------|---------|------------|-----------|---------|
| ESMFold (w/ structure recycling)    | 4.81 | 0.76 | 0.65 | 0.75 |
| ESMFold (w/ distogram recycling)  | 4.73 | 0.76 | 0.64 | 0.75 |

Hacked ESMFold to recycle the distogram instead of the structure module's output. Results are comparable on CASP14. Note that these metrics are over C-beta atoms.

| Model | RMSD | TMScore | GDT-TS | lDDT |
|----------|---------|------------|-----------|---------|
| ESMFold   | 4.57  | 0.77 | 0.67 | 0.79 |
| MiniFold (w/o recycling)   | 8.19 | 0.66 | 0.59 | 0.73 |
| MiniFold (w/ recycling)    | 5.53 | 0.74 | 0.65 | 0.77 |

Protein structure prediction results on CASP14. All results are reported using the median performance over the C-alpha coordinates. We provide the MiniFold performance with and without recycling, with 3 recycling steps considered. MiniFold nearly matches ESMFold, but only with recycling enabled

---

> ### Comment · Area_Chair_wfv7 · 2023-12-02
> **Does the response address your concerns?**
>
> @all reviewers,
>
> I would appreciate it if you could review the response and adjust your review (and rating) as necessary.
>
> AC

---

### Meta-Review · Area_Chair_wfv7 · 2023-12-10

**Metareview:**

The paper presents MiniFold, an optimized protein structure prediction model.

Reviewer 1 suggests the model's simplification and speed are its strengths but criticizes the decrease in performance and potential inaccuracies. They question the use of OpenAI's Triton over CUDA and the validity of their kernel analysis. The reviewer also suggests that the model relies too heavily on a pre-trained protein language model and does not address the chirality of the protein structure.

Reviewer 2 applauds the authors' approach and contribution, but suggests further analysis on how changes to the model affect time and performance. They question why the authors cannot directly build the 3D backbone based on the distance matrix, and how they determine the side-chain angle. They also recommend the authors to consider pruning or distilling the ESM model or using it to train a smaller model.

Reviewer 3 highlights MiniFold's efficiency but criticizes its inability to reconstruct full atoms and its lack of originality. They also lament the absence of source code for verification. They question if the authors can provide TMScore comparisons against other models and whether they have tried to reconstruct full atoms.

Reviewer 4 appreciates the clarity of the paper but critiques its limited scope, suggesting comparison with more models and evaluation on CASP proteins. They argue the paper lacks originality and needs further evidence to support its claims. The reviewer also criticizes the grandiosity of some statements and suggests the training set's composition could call into question the necessity of certain components of AlphaFold2.

**Justification For Why Not Higher Score:**

The authors have made commendable efforts to enhance the paper, which is greatly appreciated. However, several issues remain unaddressed. These include matters of originality and transferability, the model's ability to reconstruct full atoms, the inference time of the model, and the optimization of kernel fusion to minimize intermediate global memory write/read costs.

**Justification For Why Not Lower Score:**

N/A

---

### Decision · Program_Chairs · 2024-01-16

Reject